# NOTCH Signaling in Mantle Cell Lymphoma: Biological and Clinical Implications

**DOI:** 10.3390/ijms241210280

**Published:** 2023-06-17

**Authors:** Leigh Deshotels, Firas M. Safa, Nakhle S. Saba

**Affiliations:** 1Section of Hematology and Medical Oncology, Deming Department of Medicine, Tulane University, New Orleans, LA 70112, USA; 2Service d’hématologie, Centre Hospitalier du Mans, 72037 Le Mans, France

**Keywords:** mantle cell lymphoma, MCL, NOTCH1, NOTCH2, MYC

## Abstract

Despite major progress in mantle cell lymphoma (MCL) therapeutics, MCL remains a deadly disease with a median survival not exceeding four years. No single driver genetic lesion has been described to solely give rise to MCL. The hallmark translocation t(11;14)(q13;q32) requires additional genetic alterations for the malignant transformation. A short list of recurrently mutated genes including ATM, CCND1, UBR5, TP53, BIRC3, NOTCH1, NOTCH2, and TRAF2 recently emerged as contributors to the pathogenesis of MCL. Notably, NOTCH1 and NOTCH2 were found to be mutated in multiple B cell lymphomas, including 5–10% of MCL, with most of these mutations occurring within the PEST domain of the protein. The NOTCH genes play a critical role in the early and late phases of normal B cell differentiation. In MCL, mutations in the PEST domain stabilize NOTCH proteins, rendering them resistant to degradation, which subsequently results in the upregulation of genes involved in angiogenesis, cell cycle progression, and cell migration and adhesion. At the clinical level, mutated NOTCH genes are associated with aggressive features in MCL, such as the blastoid and pleomorphic variants, a shorter response to treatment, and inferior survival. In this article, we explore in detail the role of NOTCH signaling in MCL biology and the ongoing efforts toward targeted therapeutic interventions.

## 1. Introduction

Mantle cell lymphoma (MCL) is a CD5-positive, CD23-negative B cell malignancy accounting for 6% of all non-Hodgkin lymphomas (NHL) and carries a poor prognosis, with a median survival of only 4–5 years. MCL is characterized by the translocation t(11;14)(q13;q32), which results in the overexpression of the Cyclin-D1 protein [1,2,3]. Cyclin-D1 regulates the cell cycle via promoting the G1 to S transition, and its overexpression results in the uncontrolled cell growth of the malignant clone [2]. While the translocation t(11;14)(q13;q32) is a hallmark for MCL, it is not sufficient for the malignant transformation, and additional genetic insults are required [3]. MCL exhibits considerable genetic diversity when compared to other lymphomas of the B cell lineage; however, the clinical and therapeutic implications of the majority of these mutations are yet to be determined. Although there has not been a single dominant mutation identified in this disease, some of the most commonly mutated genes include ATM, CCND1, UBR5, TP53, BIRC3, NOTCH1, NOTCH2, and TRAF2 [4]. Notably, NOTCH1 and NOTCH2 were found to be mutated in multiple B cell NHLs, including 5–10% of MCL [5,6].

The NOTCH1 and NOTCH2 genes encode transmembrane receptors with distinct functions in cell development [7]. NOTCH1 expression is observed in immature T cells, and its activation directs early hematopoietic progenitors toward the T cell lineage [8]. Conversely, NOTCH2 expression is primarily seen in mature B cells and is essential for the development of splenic marginal zone B cells [9]. Both NOTCH1 and NOTCH2 regulate the expression of several proto-oncogenes, either directly or indirectly, through downstream effectors [10].

The NOTCH signaling pathway is highly conserved, and its dysregulation has been implicated in the pathogenesis of many B cell malignancies including MCL, chronic lymphocytic leukemia (CLL), splenic marginal zone lymphoma (SMZL), follicular lymphoma (FL), diffuse large B cell lymphoma (DLBCL), and multiple myeloma (MM) [11,12,13]. The mutational impact of NOTCH1 is best described in CLL as it represents one of the most frequent single gene alterations found in CLL at diagnosis (5–15% of cases) but increases to 20% in the relapsed setting [14,15]. While the exact impact of NOTCH1 mutation on CLL’s response to chemoimmunotherapy is controversial, it does not seem to impact overall outcomes in the era of targeted therapy. Nonetheless, mutations in NOTCH1 seem to co-occur more frequently with an unmutated IGHV gene [15]. The most common NOTCH1 mutation in CLL results in a premature stop codon affecting the C-terminal portion of the NOTCH1 receptor, leading to an absence of the proline (P), glutamate (E), serine (S), and threonine (T) (PEST) domain, which regulates protein stability and receptor degradation [16]. In addition to mutations in the NOTCH1 gene, CLL can amplify NOTCH1 signaling through the overexpression of the NOTCH1 receptor, taking advantage of the abundance of NOTCH1 ligands in the lymph node (LN) microenvironment [16]. While NOTCH1 pathway alterations play a critical role in CLL biology, mutations in NOTCH2 are prevalent in SMZL, affecting 20% of cases; however, like NOTCH1 mutations in CLL, the majority of NOTCH2 mutations in SMZL affect the C-terminal region PEST domain of the NOTCH2 receptor [7,11]. More recently, dysregulated NOTCH signaling has been implicated in the pathogenesis of MM as well. Several studies have demonstrated that the progression of MM from a monoclonal gammopathy of undetermined significance (MGUS) relies upon the upregulation of the NOTCH1 receptor and its ligand, JAG1 [13,17].

Mutations in both NOTCH1 and NOTCH2 have been identified in MCL and have been associated with adverse features and more aggressive MCL phenotypes such as blastoid and pleomorphic histologies [6,18,19]. In a study conducted by Bea and colleagues, NOTCH1 and NOTCH2 mutations were found in 5.2% and 4.6% of tumors, respectively [6]. A similar study by Kridel and colleagues showed that 11% of the 123 sequenced MCL samples carry a NOTCH1 mutation [18]. While NOTCH1 mutations are more commonly encountered and are similar to those observed in CLL, NOTCH2 mutations have recently been identified in MCL as an alternative and mutually exclusive occurrence [6,7].

Preclinical studies have shown that targeting NOTCH signaling pathways can be an effective therapeutic strategy in MCL. These strategies include monoclonal antibodies that block Notch ligand–receptor interactions, gamma secretase inhibitors (GSI) that block the cleavage of the NOTCH receptor and prevent its activation, or agents that target trafficking of intracellular NOTCH domain to the nucleus [5].

In this review article, we explore in detail the role of the NOTCH signaling pathway in MCL pathogenesis and the therapeutic strategies currently in development that aim to target this pathway.

## 2. Description of the NOTCH Signaling Pathway

In humans, four NOTCH genes have been described that encode four separate NOTCH receptors, NOTCH1–4. NOTCH1 and NOTCH2 are the most widely expressed in human tissues, especially at the developmental stage, while NOTCH3 is expressed in vascular smooth muscle and pericytes, and NOTCH4 is expressed in the endothelium [7].

All NOTCH receptors have a similar structure and are composed of a large NOTCH extracellular domain (NECD), a single-pass transmembrane domain (TM), and a small NOTCH intracellular domain (NICD) (Figure 1) [20]. The NECD is composed of multiple epidermal growth factor (EGF)-like repeats, followed by a negative regulatory region (NRR). The EGF-like repeats are responsible for binding to ligands expressed on neighboring cells, leading to NOTCH receptor activation. The number of EGF-like repeats varies between the NOTCH receptors: 36 for both NOTCH1 and NOTCH2, 34 for NOTCH3, and 29 for NOTCH4. The NRR is composed of three Lin12/Notch repeats (LNR) and a heterodimerization domain (HD) and functions as an inhibitor of constitutive NOTCH receptor activation in the absence of ligands (Figure 1 and Figure 2) [7,21]. The TM domain contains cleavage sites recognized by the gamma secretase complex. TM cleavage results in the release of the NICD, allowing for its translocation to the nucleus [22]. The NICD contains a protein-binding RBPJκ-associated molecule (RAM) domain, seven ankyrin repeats (ANK), a NOTCH cytokine response region (NCR), and a transcriptional activation domain (TAD) [23]. This is followed by a C-terminal region containing the PEST domain, which is an important regulator of protein stability (Figure 1) [7].

Cell-to-cell contact is a critical step in the activation of NOTCH pathways as it enables NOTCH receptors to bind to their membrane-bound ligands on neighboring cells (Figure 2) [7,10]. A total of five ligands, grouped into two families, interact with the NOTCH receptors: the Delta-like family (DLL-1, DLL-3, and DLL-4) and the Serrate family of ligands (JAG-1 and JAG-2) (Figure 2) [24]. Interaction with a ligand results in the exposure of the NOTCH receptor’s cleavage site to the protease A-Disintegrin-And-Metalloprotease (ADAM). ADAM-mediated proteolysis leads to the cleavage of the NECD portion of the NOTCH receptor. The TM domain is subsequently cleaved by a gamma secretase complex, releasing the NICD and allowing its translocation to the nucleus (a process called transactivation) [25]. Within the nucleus, The NICD binds to the transcription factor RBPJ and to a mastermind-like (MAML) transcriptional coactivator to form a trimeric complex that activates the transcription of the NOTCH target genes (Figure 2) [26,27].

NOTCH-regulated genes include HES1 and HEY1, transcription repressors that inhibit the expression of genes involved in cell differentiation and apoptosis, DTX1 and NRARP, which provide constitutive NOTCH activation by upregulating a positive feedback loop, and most importantly, MYC, which is a well-known proto-oncogene responsible for the upregulation of cell growth and metabolism [7,28,29].

There are at least two separate mechanisms in which NOTCH activity is terminated. One involves the expression of the HES and HEY family of transcriptional suppressor proteins, which are known targets of NOTCH signaling pathways. Once activated, NOTCH triggers an overexpression of HES and HEY that bind to and inhibit RBPJ, thus creating a negative feedback mechanism, thwarting further NOTCH-induced transcription [29]. Another process involves the degradation of the intracellular portion of the NOTCH receptor via the phosphorylation of the PEST domain [7]. The PEST domain serves as a recognition site for E3 ubiquitin ligases, which covalently attach ubiquitin molecules to lysine residues within the NOTCH receptor [30]. This process marks the intracellular portion of the receptor for recognition and degradation by the proteasome (Figure 2) [30].

## 3. NOTCH Pathway and Its Role in Normal B Cell Development

The NOTCH1 receptor plays an important role in the early stages of hematopoiesis, specifically in T-cell development. When Pui et al. inoculated progenitor cells transduced with Notch1-espressing retroviral vectors into the bone marrow of mice, they noticed an emergence of a large population of thymic-independent T cells in the bone marrow, in addition to a notable absence of B cell maturation [8]. These and similar findings highly suggest that NOTCH1 activation skews the progenitor cell differentiation toward T cells while suppressing the development of other cell lineages [8,31].

In bone marrow, proper B cell development requires NOTCH1 to be turned off. These antigen-naïve B cells migrate from the bone marrow to secondary lymphoid organs such as the LN and the spleen, where they undergo further maturation through transitional stages (T1 and T2) prior to becoming mature follicular B cells or marginal zone B cells [31].

The interaction between the DLL1 ligand with the NOTCH2 receptor in the marginal zone of the spleen favors marginal zone B cells over follicular B cells [31]. Conversely, Tanigeki and colleagues showed that B cells lacking the downstream transcription factor of NOTCH2, RBPJ, preferentially committed to a follicular cell lineage rather than marginal zone B cell differentiation [32]. These findings highly suggest that activated NOTCH2 in the secondary lymphoid organs is necessary for the B cell transition through the T2 stage. Following that, a continuation of NOTCH2 activation favors marginal zone B cells, while its deactivation favors follicular B cells.

On the other hand, NOTCH1 signaling regulates the differentiation of B cells into antibody-secreting cells. Santos and colleagues stimulated B cells with lipopolysaccharide in the presence of stromal cells transduced with DLL1 [33]. Although the number of B cells recovered was not increased, the antibody production was significantly increased in the DLL1-stimulated B cell population, specifically IgG and IgM [33]. In contrast, the Jagged ligand failed to reproduce these results, suggesting that there is a specificity among the NOTCH ligands with respect to antibody synthesis [34].

In addition to its effects in the spleen, NOTCH signaling plays a critical role in the development of germinal centers (GC) within the LN, where B cells rapidly proliferate and undergo somatic hypermutation, affinity maturation, and isotype switching [34]. NOTCH ligands, specifically DLL1 and JAG1, expressed by follicular dendritic cells in the germinal centers, interact with the NOTCH1 and NOTCH2 receptors expressed on GC B cells [35]. This interaction increases the survival of the GC B cells by regulating the expression of several anti-apoptotic proteins, including Bcl-2 and Mcl-1 [35].

Overall, NOTCH signaling is a critical regulator of B cell development and function, beginning early at the B cell progenitor level and continuing to the end stages of B cell differentiation and antibody production.

## 4. Cellular Effects of NOTCH Mutations and Their Roles in MCL

Mutations in NOTCH1 and NOTCH2 appear to define a more aggressive MCL subset, such as a frequent association with pleomorphic and blastoid variants and a shorter survival [6,18]. A study by Kridel et al. showed that mutated NOTCH1 represents an independent survival prognostic factor, resulting in a median overall survival (mOS) of 1.4 years compared to 3.8 years in NOTCH1 wild-type cases [18]. In a second report, none of the eight patients carrying a NOTCH2 mutation in their MCL were alive after three years compared to 62% of those without the mutation [6].

While NOTCH mutations in MCL can occur in any location, the PEST domain appears to be the most affected in both NOTCH1 and NOTCH2, harboring more than 85% of the reported mutations (Table 1) [6,18,36,37]. The most common genetic alteration in NOTCH1 is the frame shift alteration P2514Rfs*3 [6,18,36], while the majority of the reported NOTCH2 mutations occurred in exon 34, with the stop-gain R2400* substitution mutation being the most common (Table 1) [6,19]. In both cases, the mutation leads to a truncated PEST domain of the NOTCH protein, resulting in a resistance to E3 ubiquitin ligase-mediated ubiquitination and degradation and thus in a longer half-life of NICD [24].

In a recent study that aimed to establish NOTCH targets in MCL, investigators defined the two MCL cell lines, Rec-1 and SP-49, as “NOTCH-addicted” based on their high sensitivity to gamma-secretase inhibitors (GSIs) [38]. Ongoing NOTCH signaling in these cell lines is secondary to a NOTCH1 PEST domain mutation in Rec-1 and to an HLA-DMB-NOTCH4 fusion in SP-49. In the latter, the first exon of HLA-DMB is spliced into exons 24–30 of NOTCH4, resulting in a truncated NOTCH4 receptor and a lacking PEST domain. Interestingly, both Rec-1 and SP-49 showed marked reductions in Myc levels following treatment with GSIs. In comparison, GSIs failed to downregulate Myc in a Mino cell line that harbors both a NOTCH1 PEST domain mutation and a MYC-IGH rearrangement, resulting in activated NOTCH1 and MYC signaling. These findings indicate that Myc is, at least in part, under the control of NOTCH signaling. Moreover, by comparing the gene expression of NOTCH-addicted MCL cell lines to those of NOTCH1-rearranged T-ALL and breast cancer cell lines, the investigators revealed a set of MCL-specific NOTCH-regulated genes, including DTX1, LYN, and BLK (Appendix A) [38]. Interestingly, MYC was found to be activated in all NOTCH1-active cell lines; however two distinct mechanisms of activation were observed depending on the tissue type. In T-ALL, NOTCH1-dependent MYC activation occurs through an interaction between the transcription factor RBPJ and an enhancer on the 3′ side of MYC (termed T-NDME), while in MCL, it takes place at the 5′ side’s enhancers, E1 and E2 [18,38]. Other NOTCH1 targets in MCL are genes involved in cytokine and interleukin signaling, such as IL6R, IL10RA, IL21R, and the regulation of B cell receptor activation, including Src-family kinases (BLK, FYN, FGR, and LYN), adaptor proteins (PIK3AP, BLNK, NEDD9, and SH2B2), BCR signaling modulators (CD21 and CD300A), and members of the Fc-receptor-like family (FCRL3, FCRL4, and FCRL5) [38].

A second study evaluated the effects of NOTCH1 activation on MCL biology following incubation with DLL4 in the MCL cell lines Mino and Jeko-1 [5]. DLL4 resulted in a significant increase in cellular migration, which subsequently was abrogated by the anti-NOTCH antibody OMP-52M51. Investigators simultaneously performed a human umbilical vein endothelial cells (HUVEC) tube formation assay to evaluate the effect of NOTCH1 stimulation on angiogenesis. They showed an increase in the number of branch points promoted by the supernatants of the DLL4-stimulated Mino and Jeko-1, indicating an induction of angiogenesis, which was once again counteracted by OMP-52M51 [5]. Cell stimulation with DLL4 also led to the phosphorylation of ERK and MEK, indicating an activation of the BCR pathway [5].

To interrogate the signaling pathways activated in MCL in vivo, we contrasted gene expression profiles of tumor samples isolated from the blood and LNs of previously untreated patients with active disease [37]. Interestingly, NOTCH signaling was among the top activated pathways in LN-resident MCL cells when compared to blood. In the absence of NOTCH mutations in the tested LN samples, our findings suggest that mutation-independent NOTCH signaling is ongoing in MCL in vivo and is preferentially located in the LN [37]. In addition to NOTCH pathway activation, BCR signaling and MYC-regulated genes were also detected as most active in the LN, further implicating a relationship between the NOTCH, BCR, and MYC pathways [37].

We have recently established a NOTCH2 R2400* mutated MCL cell line (Arbo) derived from a patient with blastoid MCL [19]. R2400* is located in the PEST domain and results in increased protein stability, as confirmed by NOTCH2 overexpression in Arbo to other MCL cell lines. Th dependence of the Arbo cell line on NOTCH2 signaling for survival was evidenced by its high sensitivity to gliotoxin, a NOTCH2 transactivation inhibitor. Moreover, Arbo was sensitive to ibrutinib and resistant to venetoclax, further underscoring the interplay between NOTCH2 and BCR signaling and potentially implicating NOTCH2 in venetoclax resistance [19].

These studies provide significant insights into the role of NOTCH signaling in the pathobiology of MCL and underscore a possible impact on major survival pathways such as BCR signaling, MYC-dependent, and apoptotic pathways.

## 5. Inhibition of NOTCH Pathways

The inhibition of NOTCH signaling pathways in cancer has proven to be a difficult task, owing in part to the complex mechanism of action of these receptors. Multiple steps are involved in pathway activation, from binding to the ligands to the cleavage of the NICD by gamma secretases, in addition to the effects of NOTCH mutations that induce a ligand-independent pathway activation. Nonetheless, multiple strategies to target NOTCH signaling pathways in cancer have been developed. These strategies include antibodies blocking either extracellular NOTCH domain or its ligands, inhibitors of gamma secretase, decoy agents that bind to the NOTCH receptor, preventing ligand-dependent activation, and agents that target NICD trafficking from the cell membrane to the nucleus [39,40]. Testing of these strategies is ongoing, mostly at the preclinical stage, in a variety of lymphoid cancers, including MCL, CLL, MZL, and T-ALL [41,42,43,44,45]. Here, we will discuss the NOTCH inhibitors tested in MCL.

### 5.1. NOTCH-Directed Monoclonal Antibodies

The only NOTCH-directed antibody studied in MCL is the humanized anti-NOTCH1 antibody OMP-52M51 (brontictuzumab), a full-length IgG2 humanized monoclonal antibody directed against the LNR and HD domains of NOTCH1. Since brontictuzumab inhibits signaling downstream to NOTCH1 by blocking ligand-dependent receptor activation, its activity is not affected by the presence of PEST domain mutations [46]. The in vitro treatment of DLL4-stimulated Mino cells with brontictuzumab inhibited NOTCH1 activation and abrogated angiogenesis, DNA damage repair, and cellular migration and adhesion. Although brontictuzumab decreased the level of cleaved NOTCH1 in murine MCL xenografts, there was no effect on tumor cell counts and viability [5].

Following encouraging in vitro data, brontictuzumab was tested in a phase 1 clinical trial in 24 patients with relapsed or refractory hematologic malignancies, including 4 patients with MCL [47]. The antibody was relatively well tolerated, with dose-limiting toxicity occurring in two patients. Diarrhea was the most common adverse event, occurring in about one-fifth of patients, followed by fatigue and anemia. Unfortunately, brontictuzumab showed mediocre clinical effects, with only one partial response in a patient with transformed mycosis fungoides and two stable diseases in two others (MCL and transformed mycosis fungoides). Among the three patients who had activating PEST domain mutations in NOTCH1, only one achieved stable disease as a best response [47].

### 5.2. Gamma Secretase Inhibitors

Gamma secretase inhibitors (GSI) were initially developed for use in Alzheimer’s disease, given the role of gamma secretase in the cleavage of amyloid precursor protein, a key element in disease progression [48,49]. Following discoveries of gamma secretase’s role in cancer, GSIs were repurposed for testing in both solid and hematological malignancies [43,50]. In the preclinical testing of a small-molecule GSI named Compound E on MCL, researchers showed that Compound E significantly decreased Rec-1 and SP-49 cell viabilities at low micromolar concentrations with a concomitant decrease in NICD levels, reflecting a successful inhibitory effect on NOTCH1 cleavage [38].

Among GSIs, the most extensively studied in the clinical sphere to date is RO4929097, a small-molecule inhibitor of gamma secretase. RO4929097 has been tested in phase I and II trials on various solid tumors, including melanoma, glioma, lung, breast, and pancreatic malignancies, and resulted in only a modest clinical benefit [23,51,52,53,54,55]. As an example, when tested in a phase II clinical trial in patients with previously treated metastatic pancreatic cancer, RO4929097 as a single agent failed to induce any objective responses, with only three out of twelve patients achieving a stable disease as the best clinical response, resulting in a median progression-free survival of 1.5 months [56]. A similar phase II of RO4929097 in ovarian cancer resulted only in stable disease as the best clinical response in one-third of the forty-five treated patients and a median progression-free survival of 1.3 months [57]. In both studies, RO4929097 was well tolerated, with fatigue and nausea being the most commonly reported adverse events, while anemia and diarrhea were the most common grade 3 and 4 toxicities, occurring in less than 5% of patients [56,57]. There are currently no clinical trials evaluating GSI in B cell lymphoma.

### 5.3. Inhibition of NOTCH Transactivation

Gliotoxin is an Aspergillum-derived secondary metabolite that was found to be a potent inhibitor of NOTCH2 transactivation. Hubmann and colleagues showed that gliotoxin exerts an anti-leukemic effect on CLL by inducing apoptosis at the nanomolar range in patient-derived samples, irrespective of the IGHV mutational status or cytogenetic anomalies [45]. The effect of gliotoxin on NOTCH2 signaling resulted in decreased CD23 expression, indicating a suppressive effect on the CD23-encoding gene FCER2, a known target of NOTCH2 [45].

The uniform expression of CD23 in CLL contrasts with its absence in MCL, a pattern used to differentiate between these two CD5-expressing B cell malignancies. Nonetheless, a subset of MCL expresses CD23 with ill-defined causes and consequences. In keeping with the control of NOTCH2 over CD23 expression, our NOTCH2 R2400* mutant MCL cell line Arbo showed a strong CD23 expression which was downregulated by gliotoxin in a dose-dependent manner. Additionally, Arbo cells were highly sensitive to gliotoxin, with an IC_50_ of 100 nM, while the NOTCH2 wild-type MCL cell lines Jeko-1 and UPN-1 were resistant [19]. To date, there are no clinical trials evaluating gliotoxin in lymphoid malignancies.

Although not tested in MCL, additional ways of inhibiting NOTCH pathways in the preclinical setting include targeting NOTCH degradation and the ICN1 complex. These strategies hold the promise of being more selective to the target, which could translate into improved efficacy and safety profiles [58,59,60,61].

## 6. Conclusions

The role of NOTCH pathways in MCL biology is becoming more convincing than ever. Both the NOTCH1 and NOTCH2 pathways seem to be involved, perhaps through activating mutations and possibly through a mutation-independent pathway activation in the LN microenvironment. A direct effect on MYC is evident, and a strong association with BCR and apoptotic pathways is likely. Unfortunately, the therapeutic applicability of the available NOTCH inhibitors in humans remains unsuccessful, owing in part to the heterogeneity of MCL and the complexity of the NOTCH pathways in this disease. Despite current challenges in the therapeutic use of NOTCH inhibitors, further understanding of NOTCH signaling may offer novel options for patients with MCL. Moving forward, incorporating the powerful tool of drug invention, in addition to novel discoveries that could redefine NOTCH signaling pathobiology, will reveal the most appropriate means of targeting NOTCH signaling and will pave the road to a tailored, clinically effective combination therapy.

## Figures and Tables

**Figure 1 ijms-24-10280-f001:**
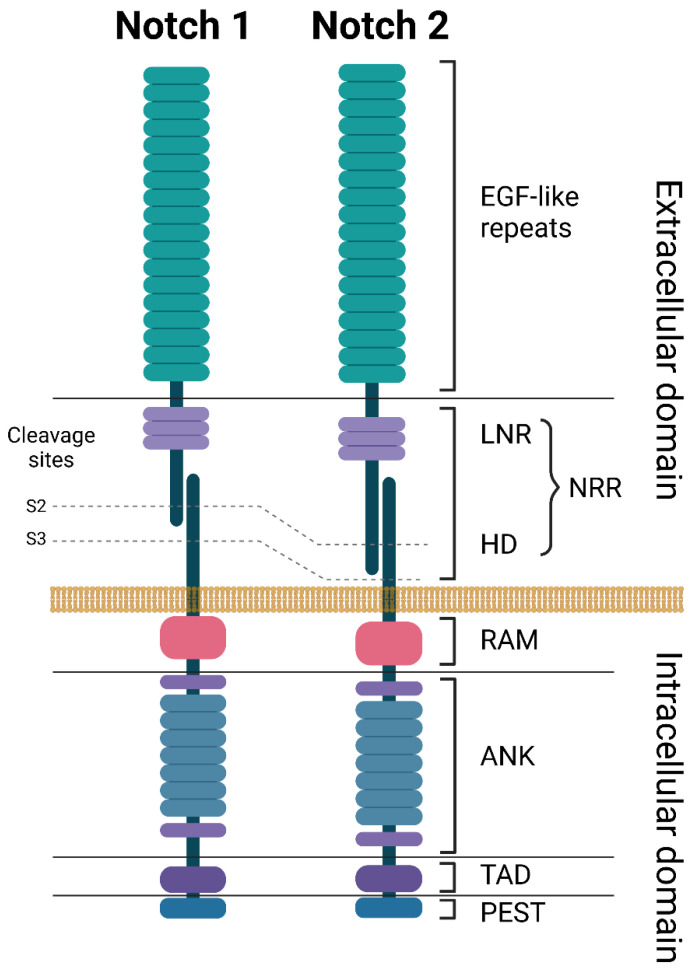
Structures of NOTCH1 and NOTCH2 receptors. ANK: ankyrin repeats, EGF: epidermal growth factor, HD: heterodimerization domain, LNR: Lin-12 Notch repeats, NRR: negative regulatory region, PEST: proline (P), glutamate (E), serine (S), and threonine (T) domain, RAM: RBP-Jkappa-associated module, TAD: transcriptional activation domain. Created with BioRender.com. accessed on 5 June 2023.

**Figure 2 ijms-24-10280-f002:**
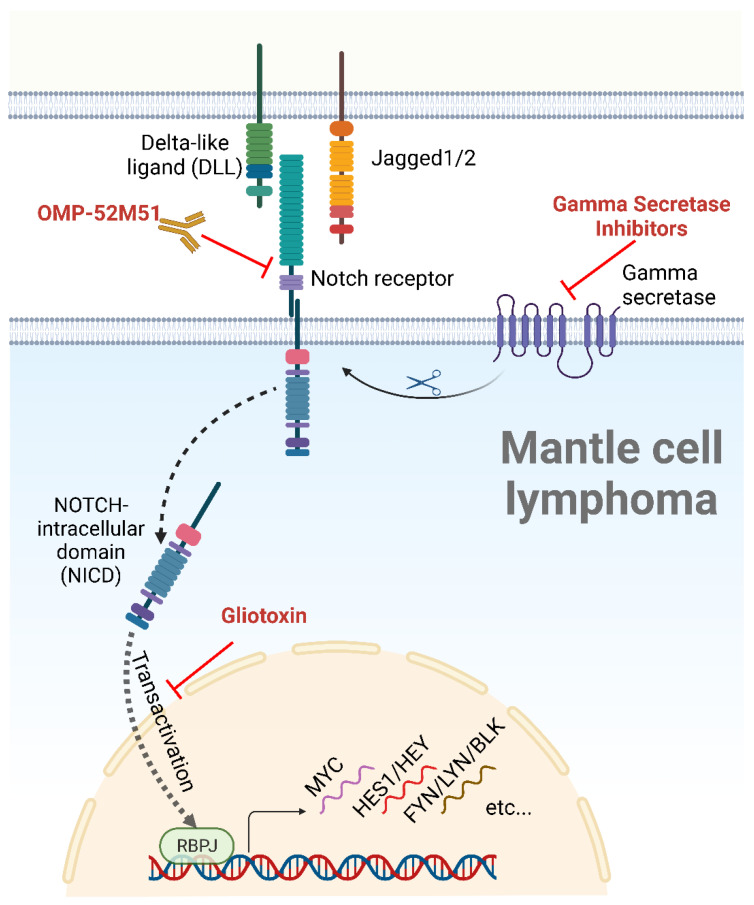
The NOTCH signaling pathway, ligands, and inhibitors. After binding to a cellular ligand, the NOTCH receptor is cleaved by a gamma secretase, allowing the nuclear translocation of the NICD, a process known as transactivation. Following nuclear translocation, the NICD induces the transcription of NOTCH target genes through its interaction with the DNA-binding partner RBPJ. NOTCH inhibitors tested in MCL are depicted in red. Created with BioRender.com, accessed on 5 June 2023.

**Table 1 ijms-24-10280-t001:** Common NOTCH1 and NOTCH2 mutations in MCL.

	NOTCH1			NOTCH2		
	Mutation	Confirmed Somatic	% of NOTCH1 Mutations †	Mutation	Confirmed Somatic	% of NOTCH2 Mutations †
EGF like repeats	D259N [36]	No	4	R91L [37]	Yes	6.6
	P915R [36]	No	4	T197I [37]	No	6.6
	G1088S [18,36]	No	4	G225R [37]	Unknown	6.6
ANK	R1608H [18]	No	4			
	D2108N [36]	Yes	4			
PEST	H2428Pfs*7 [18]	Yes	4	Q2285* [6]	Yes	6.6
	G2281fs*72 [6]	Unknown	4	K2292fs*20 [6]	Unknown	6.6
	R2431Pfs*5 [18,36]	No	4	H2293fs*2 [6]	Unknown	6.6
	Q2444* [18]	Yes	4	P2359A [37]	No	6.6
	E2460* [18,36]	Yes	4	Q2360* [6]	Yes	6.6
	Q2487* [18]	Yes	4	S2391fs*2 [6]	Unknown	6.6
	V2504fs*3 [6]	Unknown	4	R2400* [6,19,37]	Yes	40
	H2507Pfs*9 [18]	No	4			
	P2514Rfs*4 [6,18,36]	Yes	56			

† The frequency of each mutation was calculated by dividing the number of each individual mutation by the total number of reported NOTCH mutations (NOTCH1: 25 mutations in 24 samples; NOTCH2: 14 mutations in 14 samples). One sample carried two NOTCH1 mutations (D259N and P2514Rfs*4) [36], and one sample carried a NOTCH1 (P2514Rfs*4) and a NOTCH2 (K2292fs*20) mutation [6].

## Data Availability

Publicly available datasets were analyzed in this study. This data can be found in Ref. [37] and are available in GEO under the accession number GSE70926.

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
