# Peer review of "NOTCH Signaling in Mantle Cell Lymphoma: Biological and Clinical Implications"

_ijms, 2023, doi:10.3390/ijms241210280_

Round 1
Reviewer 1 Report
In this manuscript, Deshotels et al review NOTCH signaling in MCL. NOTCH signaling is a critical regulator of B cell development and function, and the authors explore in detail the role of the pathway in MCL pathogenesis and the therapeutic strategies currently in development: NOTCH directed monoclonal antibodies, gamma secretase inhibitors, and inhibitors of NOTCH transactivation. The paper is well-written and referenced and covers most of the current data on a relevant topic.
Author Response
REVIEWER 1
- In this manuscript, Deshotels et al review NOTCH signaling in MCL. NOTCH signaling is a critical regulator of B cell development and function, and the authors explore in detail the role of the pathway in MCL pathogenesis and the therapeutic strategies currently in development: NOTCH directed monoclonal antibodies, gamma secretase inhibitors, and inhibitors of NOTCH transactivation. The paper is well-written and referenced and covers most of the current data on a relevant topic.
Response: We appreciate the reviewer’s time to review our manuscript and for providing valuable and positive feedback.
Reviewer 2 Report
Deshotels et.al reviewed NOTCH signaling in mantle cell lymphoma, with well explanation in both biological and clinical perspective. I don't have any negative comments on this manuscript at all and fully support it to be published.
Author Response
- Deshotels et.al reviewed NOTCH signaling in mantle cell lymphoma, with well explanation in both biological and clinical perspective. I don't have any negative comments on this manuscript at all and fully support it to be published.
Response: We appreciate the reviewer’s time to review our manuscript and for providing valuable and positive feedback.
Reviewer 3 Report
The authors provided a rather comprehensive review on NOTCH signaling in MCL. The topic is interesting and should attract a broad range of readership. For the most part, the manuscript was reasonably well written, however, there are several issued that need to be addressed by the authors:
1. Please refer to MCL morbidity and mortality statistics
2. Add future direction in the conclusion section
3. Please add more explanations in the legend of the figures
Author Response
- Please refer to MCL morbidity and mortality statistics.
Response: We agree with the reviewer and have added the following comment on the incidence and overall survival on MCL in the first paragraph of the introduction: “MCL accounts for 6% of all non-Hodgkin lymphomas and carries an overall survival of 4-5 years [1-3].”
- Add future direction in the conclusion section.
Response: Following the reviewer’s recommendation, we have added the following comment to the conclusion section: “Moving forward, incorporating the powerful tool of drug invention, along with novel discoveries that could redefine NOTCH signaling pathobiology, will reveal the most appropriate way of targeting NOTCH signaling and pave the road to a tailored, clinically effective combination therapy.”
- Please add more explanations in the legend of the figures.
Response: More explanations were added to the figures legends as recommended by the reviewer.
Reviewer 4 Report
This is a nice, review. It is informative but concise. Notch is an important driver in lymphoma and other cancers and thus the review is pertinent. I have some minor criticisms and comments, below
1.- Page 2. paragraphh 3: "Mutations in both NOTCH1 and NOTCH2 have been identified in MCL...". >>They should provide information so as the frequency of NOTCH1 and NOTCH2 in MCL.
2.- Page 3, last paragraph: "The number of repeats varies between the NOTCH receptors and ranges from 29 to 36" >>They may provide here or in the figure informaiton on the different number of repeats dependneing on the Notch receptor
3.- Figure 2: NICD is not labelled in the figure
4.- Page 2, 2nd paragraph: . "NOTCH signaling and inhibit further NOTCH activation [29]" > They should briefly inform on the mechanism by which HES and HEY inhibits Notch activity
5.- Page 5 “Mutations in NOTCH1 and NOTCH2 occur in 5 to 10% of MCL” >> Same % fior both genes?
6.- Table 1: It would be desirable to give an idea of which of domains is more mutated or the frequency of the mutations. In the text it only stated that the R2428Pfs5 is the most frequent alteration
7.- Page 6: It would be helpful a table thar summarizes and organized the Notch target genes in MCL
MINOR:
Page 3, 3rd line: RBPJk-assocoated molecule..:" The mean here DOMAIN
Page 3, 6th line: “: ...PEST domain·”. Indicate what does it stands or: ich in proline (P), glutamic acid (E), serine (S), and threonine (T).
Page 3 last paragraph: The official HUGO names are JAG1 and JAG2
Page 6: “While NOTCH1-dependent MYC activation in T-ALL occurs through an interaction between the transcription factor …” >> There is a problem with this sentence. "While" does not sound well here.
Page 7: “Dependence of Arbo on NOTCH2….” >>> Arbo cells is more appropriate
Author Response
MAJOR
- Page 2. paragraph 3: "Mutations in both NOTCH1 and NOTCH2 have been identified in MCL...". >>They should provide information so as the frequency of NOTCH1 and NOTCH2 in MCL.
Response: We agree with the reviewer’s suggestion, and we have added the following two sentences which summarize the current data, “In this study, NOTCH1 and NOTCH2 mutations were found in 5.2% and 4.6% of 172 analyzed MCL primary samples, respectively [6]. A similar study by Kridel and colleagues showed that 11% of the 123 sequenced MCL samples carry a NOTCH1 mutation [18].”
- Page 3, last paragraph: "The number of repeats varies between the NOTCH receptors and ranges from 29 to 36" >>They may provide here or in the figure information on the different number of repeats depending on the Notch receptor.
Response: We thank the reviewer for this comment, and accordingly, we have adjusted the sentence on page 3 to reflect the specific number of EGF-like repeats on each of the NOTCH receptors, as follows: “Both NOTCH1 and NOTCH2 contain 36 EGF-like repeats, whereas NOTCH3 contains 34 EGF-like repeats and NOTCH4 contains 29 EGF-like repeats [7].”
- Figure 2: NICD is not labelled in the figure.
Response: We have adjusted the identification of the intracellular domain to read as “NOTCH-intracellular domain (NICD)” on figure 2.
- Page 2, 2nd paragraph: "NOTCH signaling and inhibit further NOTCH activation [29]" > They should briefly inform on the mechanism by which HES and HEY inhibits Notch activity.
Response: We have clarified the mechanism by which HES and HEY inhibit NOTCH activation, which is through a negative feedback loop, as follows: “One involves the expression of the HES and HEY family of transcriptional suppressor proteins, which are known targets of NOTCH signaling pathways. Once activated, NOTCH triggers an overexpression of HES and HEY that bind to and inhibit RBPJ, thus creating a negative feedback mechanism thwarting further NOTCH-induced transcription” [29].
- Page 5 “Mutations in NOTCH1 and NOTCH2 occur in 5 to 10% of MCL” >> Same % for both genes?
Response: The frequency of NOTCH1 and NOTH2 mutations were stated in the introduction and detailed in the updated table 1 (see#6 below). To minimize confusion, we deleted “occur in 5 to 10% of” from the sentence referred to by the reviewer.
- Table 1: It would be desirable to give an idea of which of domains is more mutated or the frequency of the mutations. In the text it only stated that the R2428Pfs5 is the most frequent alteration
Response:
We appreciate the reviewer’s suggestion and added the frequency of each individual mutation to table 1. The frequency of each mutation was calculated by dividing the number of each individual mutation by the total number of reported NOTCH mutations.
- Page 6: It would be helpful to have a table that summarizes and organizes the Notch target genes in MCL
Response:
We included four gene signatures (two up- and two down-) regulated by NOTCH signaling in MCL and T-ALL, in supplementary table S1. Those signatures were the only ones reported in lymphoid malignancies in the literature thus far.
MINOR:
- Page 3, 3rd line: RBPJk-associated molecule..:" The mean here DOMAIN.
Response: We have adjusted the sentence accordingly.
- Page 3, 6th line: “: ...PEST domain·”. Indicate what does it stands or: ich in proline (P), glutamic acid (E), serine (S), and threonine (T).
Response: Please see prior to this on page 2, paragraph 2, where we first identify and define the PEST domain and the abbreviation.
- Page 3 last paragraph: The official HUGO names are JAG1 and JAG2.
Response: We have adjusted the names of these ligands on this page, as well as other areas where we addressed the JAG1 and JAG2 ligands.
- Page 6: “While NOTCH1-dependent MYC activation in T-ALL occurs through an interaction between the transcription factor …” >> There is a problem with this sentence. "While" does not sound well here.
Response: We agree with the reviewer that this sentence could use clarification and we have changed it to read as, “In T-ALL, NOTCH1-dependent MYC activation occurs through an interaction between the transcription factor RBPJ and an enhancer on the 3’ side of MYC (termed T-NDME), while in MCL it takes place at the 5’ side’s enhancers E1 and E2”.
- Page 7: “Dependence of Arbo on NOTCH2….” >>> Arbo cells is more appropriate.
Response: We changed the sentence to read as, “Dependence of the Arbo cell line on NOTCH2 signaling…”